# Complex Dynamics in Simple Neural Networks: Understanding Gradient Flow in Phase Retrieval

**Stefano Sarao Mannelli**[1], **Giulio Biroli**[2], **Chiara Cammarota**[3,4],
**Florent Krzakala**[5], **Pierfrancesco Urbani**[1], **and Lenka Zdeborová**[6]

## Abstract

Despite the widespread use of gradient-based algorithms for optimizing high-dimensional non-convex functions, understanding their ability of finding good minima instead of being trapped in spurious ones remains to a large extent an open problem. Here we focus on gradient flow dynamics for phase retrieval from random measurements. When the ratio of the number of measurements over the input dimension is small the dynamics remains trapped in spurious minima with large basins of attraction. We find analytically that above a critical ratio those critical points become unstable developing a negative direction toward the signal. By numerical experiments we show that in this regime the gradient flow algorithm is not trapped; it drifts away from the spurious critical points along the unstable direction and succeeds in finding the global minimum. Using tools from statistical physics we characterize this phenomenon, which is related to a BBP-type transition in the Hessian of the spurious minima.

## 1   Introduction

In many machine learning applications one optimizes a non-convex loss function; this is often achieved using simple descending algorithms such as gradient descent or its stochastic variations. The positive results obtained in practice are often hard to justify from the theoretical point of view, and this apparent contradiction between non-convex landscapes and good performance of simple algorithms is a recurrent problem in machine learning.

A successful line of research has studied the geometrical properties of the loss landscape, distinguishing between good minima - that lead to good generalization error - and spurious minima - associated with bad generalization error. The results showed that in some regimes, for several problems from matrix completion [1] to wide neural networks [2, 3], spurious minima disappear and consequently under weak assumptions [4] gradient descent will converge to good minima. However, these results do not justify numerous other results showing that good and spurious minima are present, but systematically gradient descent works [5, 6]. In [7] it was theoretically shown that in a toy model - the spiked matrix-tensor model - it is possible to find good minima with high probability in a regime where exponentially many spurious minima are provably present. In [8] it was shown that this is due to the presence of the so-called threshold states in the landscape, that play a key role in the

1. Université Paris-Saclay, CNRS, CEA, Institut de physique théorique, 91191, Gif-sur-Yvette,France.
2. Laboratoire de Physique de l'Ecole normale supérieure ENS, Université PSL, CNRS, Sorbonne Université, Université Paris-Diderot, Sorbonne Paris Cité Paris, France
3. Dipartimento di Fisica, Sapienza Università di Roma, P.le A. Moro 5, 00185 Rome, Italy
4. Department of Mathematics, King's College London, Strand London WC2R 2LS, UK
5. IdePHICS laboratory, EPFL, Switzerland
6. SPOC laboratory, EPFL, Switzerland

dynamics of the gradient flow [9, 10]: at first attracting it, and successively triggering the converge towards lower minima under certain conditions [11, 12]. However, the spiked matrix-tensor model is an unsupervised learning model and it remained open whether the picture put forward in [7, 8] happens also in learning with neural networks.

In this work we thus study learning with a simple single-layer neural network on data stemming from the well-known phase retrieval problem - that consists of reconstructing a hidden vector having access only to the absolute value of its projection onto random directions. The problem emerges naturally in a variety of imaging applications where the intensity is easier to access than the phase [13–16] but it appears also in acoustics [17] and quantum mechanics [18]. The phase retrieval problem considered here leads to a high-dimensional and non-convex optimization problem defined as follows.

**Phase retrieval.** Consider $\alpha N$ $N$-dimensional sensing vectors $\boldsymbol{x}_m$ with unitary norm and generated according to a centered Gaussian distribution, and the true labels $y_m^2$ with $y_m = \langle \boldsymbol{x}_m, \boldsymbol{W}^* \rangle$ and $\boldsymbol{W}^*$ an unknown teacher-vector (the signal in the phase retrieval literature) from $\mathbb{S}^{N-1}(\sqrt{N})$. Given the dataset, the goal is to build an estimator $\boldsymbol{W}$ of $\boldsymbol{W}^*$ by minimizing the loss function

$$\mathcal{L}(\boldsymbol{W}) = \frac{1}{2} \sum_{m=1}^{\alpha N} \left( \langle \boldsymbol{x}_m, \boldsymbol{W} \rangle^2 - \langle \boldsymbol{x}_m, \boldsymbol{W}^* \rangle^2 \right)^2 = \frac{1}{2} \sum_{m=1}^{\alpha N} \ell(\hat{y}_m, y_m), \tag{1}$$

with $\hat{y}_m = \langle \boldsymbol{x}_m, \boldsymbol{W} \rangle$ and $\ell(\hat{y}, y) = (\hat{y}^2 - y^2)^2$ is a modified square loss commonly used in the literature [19, 20] that ensures a smoother landscape compared to the square loss with the absolute values. The loss function, Eq. (1), is minimized using gradient-descent flow on the sphere starting from random initialization

$$\begin{aligned} \dot{\boldsymbol{W}}(t) &= -\nabla_{\boldsymbol{W}} \mathcal{L}(t) + \mu(t) \boldsymbol{W}(t), \\ W_i(t=0) &\sim \mathcal{N}(0,1) \quad \forall\, i, \end{aligned} \tag{2}$$

with $\mu(t)$ the Lagrange multiplier that enforces the spherical constraint during the dynamics. The value of $\mu(t)$ can be readily evaluated by taking the scalar product of the gradient of the loss with $\boldsymbol{W}$ and dividing by $N$

$$\mu(t) = \frac{1}{2N} \sum_{m=1}^{\alpha N} \frac{\partial}{\partial \hat{y}_m} \ell(\hat{y}_m(t), y_m) \hat{y}_m(t). \tag{3}$$

We can finally define the Hessian of the problem, denoting $\delta_{i,j}$ the Kronecker delta, it reads

$$\mathcal{H}_{i,j}(\boldsymbol{W}) = \frac{1}{2} \sum_{m=1}^{\alpha N} \frac{\partial^2}{\partial \hat{y}_m^2} \ell(\hat{y}_m, y_m) x_{m,i} x_{m,j} - \mu \delta_{i,j}. \tag{4}$$

**Related work and our main contributions.**

Numerous algorithms can be applied to achieve a good reconstruction of the hidden signal (teacher vector) [19, 21]. For random i.i.d. data and labels generated by a teacher the information-theoretically optimal generalization error has been analyzed in [22], showing that for $\alpha < \alpha_{\mathrm{WR}} = 0.5$ in the limit of $N \to \infty$ no estimator is able to obtain a generalization error better than a random guess. On the other hand for $\alpha > \alpha_{\mathrm{IT}} = 1$ algorithms (not necessarily polynomial ones) exist that are able to achieve zero generalization error. While the weak-recovery threshold is achievable with efficient algorithms [23], the information-theoretic threshold is conjectured not to be and an algorithmic gap has been conjectured to exist, with perfect recovery achievable with the approximate message passing algorithm only for $\alpha > \alpha_{\mathrm{alg}} = 1.13$ [22].

In the present paper we are interested in the algorithmic performance of the gradient descent algorithm. The motivation of the present work is not to compare vanilla gradient descent to other algorithms but rather use this well studied phase-retrieval problem to get insights on the properties of the gradient descent algorithm in high-dimensional non-convex optimization. The spurious minima are known to disappear in phase retrieval if the number of samples in the dataset scales as $O(N \log^3 N)$ with $N$ the input dimension [24]. Later [20] showed that randomly initialized plain gradient descent will solve the phase retrieval problem with $O(N \operatorname{poly}(\log N))$ samples. An open theoretical question is whether randomly initialized gradient descent is able to solve the phase retrieval problem with $O(N)$ samples. We explore this question in the present work.

Our main contributions can be summarized in the following points:

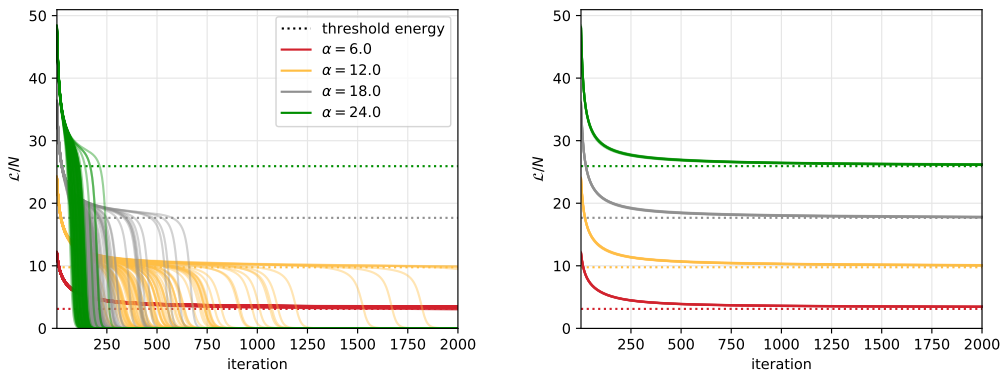

Figure 1: Evolution of the training loss for systems of size $N = 16384$ with number of samples $\alpha N$. The left panel has labels created using a teacher, while the right panel has random labels constructed using a Gaussian distribution with variance matching the teacher-labels. The left figure shows that also the simulations with a teacher approach the threshold energy before transiting to the global minimum, if $\alpha$ is large enough.

- We show empirically that in the phase retrieval problem, randomly initialized gradient flow is attracted by the so-called threshold states. We show that as the number of samples increases, the geometry of the threshold states changes from minima to saddles with the slope pointing toward the teacher-vector. This transition is akin to the BBP phase transition known in random matrix theory [25]. This transition affects gradient descent that, following the slope, achieves zero generalization error.

- We obtain a close formula for the number of samples per dimension $\alpha_c$ at which this transition happens. This depends on the joint probability distribution of the labels of the student and the teacher at the threshold.

- Using the replica theory from statistical physics as a non-rigorous proxy we characterize the approximate distribution of the labels of the student and the teacher leading to a approximate prediction for threshold $\alpha_c = 13.8$, notably suggesting that the additional logarithmic factors in the previous works might not be needed.

## 2 BBP on the threshold states

In Fig. 1 (left) we show the loss as a function of iteration time for the phase retrieval problem, defined above, with varying number of samples to dimension ratio $\alpha$ in many different runs (full lines). In the right hand side of the figure we show the loss but this time for labels that do not come from a teacher network, but that are randomly reshuffled. We see that in that case the loss converges after a very long time towards a value marked by the dotted line (reproduced also in the left part), that we defined to be the so-called threshold energy. We see that for small $\alpha$, e.g. $\alpha = 6$ the train loss on the phase retrieval problem does not decrease to zero (nor the test one), while for the larger values $\alpha = 18$ and $24$ it does very rapidly. The value $\alpha = 12$ is close to a critical regime where some realization find perfect generalization and other do not, with the dynamics staying for a long time close to the threshold energy.

In a different model, the spiked matrix-tensor, Ref. [8] described exactly this phenomenology and showed that gradient flow starting from random initial conditions has a transition when the Hessian of the spurious minima that trap the dynamics, the so-called threshold states, display a BBP transition [25]. This leads to the emergence of a descending direction toward the informative minimum which is correlated with the ground truth signal $\boldsymbol{W}^*$. In Fig. 2 we argue that the same mechanism is at play in the phase retrieval problem and based on these insights we derive an analytic equation for the corresponding threshold in Sec. 2.1.

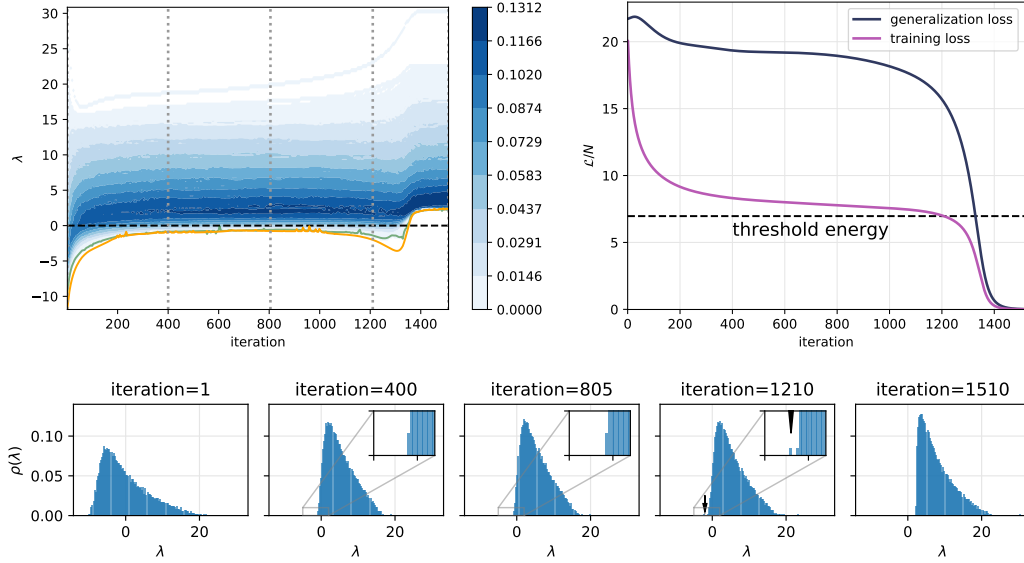

Figure 2: Properties of the Hessian for phase retrieval of a system of size $N = 2048$ at $\alpha = 10$. On the left figure, we show the evolution of the density of the bulk of the eigenvalues, from zero density in white to high density in blue, the smallest eigenvalue in orange, and the second smallest in green. The picture shows that a BBP transition occurs when the training loss approaches the threshold energy. The right panel depicts the evolution of the training loss (purple) and the generalization loss (dark blue) in time. The training loss rapidly approaches a plateau at the level of the threshold states (black dashed line) and converges towards the teacher after the BBP transition. Namely, when the smallest eigenvalue detaches from the rest of the bulk. The bottom panels show the distribution of the eigenvalues at five different instants in the dynamics. At iteration $\approx 1210$ we cross the threshold energy and we observe an isolated eigenvalue detaching from the bulk.

## 2.1   Theory for the BBP threshold

Based on the numerical results just presented, we aim to obtain an equation determining the value of threshold $\alpha_c$ such that for $\alpha > \alpha_c$ the BBP transition occurs, whereas for $\alpha < \alpha_c$ it does not. For $\alpha < \alpha_c$ the system is at long times trapped in the threshold states and not able to recover, even weakly, the signal. We define $P(\hat{y}, y)$ the long-time limit of the distribution of the estimated labels and the true labels; $P(\hat{y}, y)$ allows us to study the Hessian of the threshold states, which is the random matrix defined in (4) with $\hat{y}_\mu$ and $y_\mu$ distributed following the law $P(\hat{y}, y)$.

A type of random matrix $\mathcal{M}_{i,j}$ with similar structure as the contribution to the Hessian coming from the Loss function, $\mathcal{H}_{i,j} = -\mathcal{M}_{i,j} - \mu\delta_{i,j}$, has been studied recently in [26] and the convergence in probability for large $N$ of the largest and the second largest eigenvalues of such matrix was proven. We applied the results of the Theorem 1 of [26] to determine the behavior of the smallest and second smallest eigenvalue of the Hessian. Call

$$\Psi_\alpha(\lambda) = \lambda\left[\frac{1}{\alpha} - \mathbb{E}_{\hat{y},y}\left(\frac{\alpha\partial_{\hat{y}}^2\ell(\hat{y},y)}{2\lambda + \alpha\partial_{\hat{y}}^2\ell(\hat{y},y)}\right)\right], \qquad \Phi(\lambda) = -\lambda\mathbb{E}_{\hat{y},y}\left(\frac{\alpha\partial_{\hat{y}}^2\ell(\hat{y},y)y^2}{2\lambda + \alpha\partial_{\hat{y}}^2\ell(\hat{y},y)}\right); \quad (5)$$

let $\bar{\lambda} = \arg\min \Psi_\alpha(\lambda)$ and $\xi_\alpha(\lambda) = \Psi_\alpha(\max\{\lambda, \bar{\lambda}\})$ then the largest eigenvalue of $\mathcal{M}_{i,j}$ is $\lambda_1 \to_{\mathbb{P}} \xi_\alpha(\lambda_\alpha^*)$ with $\lambda_\alpha^*$ being the solution of $\xi_\alpha(\lambda) = \Phi(\lambda)$. The second largest is $\lambda_2 \to_{\mathbb{P}} \Psi_\alpha(\bar{\lambda}_\alpha)$.

A BBP transition occurs at $\alpha_{\mathrm{BBP}}$, the largest $\alpha$ such that $\lambda_1 = \lambda_2$. Following [26] this leads to an equation on $\bar{\lambda}$ and $\alpha_{\mathrm{BBP}}$ which reads:

$$\frac{1}{\alpha_{\mathrm{BBP}}} = \mathbb{E}_{\hat{y},y}\left(\frac{\alpha_{\mathrm{BBP}}\partial_{\hat{y}}^2\ell(\hat{y},y)}{2\bar{\lambda} + \alpha_{\mathrm{BBP}}\partial_{\hat{y}}^2\ell(\hat{y},y)}\right)^2. \tag{6}$$

We now use an additional assumption, which comes from studies of gradient descent dynamics of mean-field spin-glasses [27], that the threshold states are marginal, i.e. the smallest eigenvalue of their

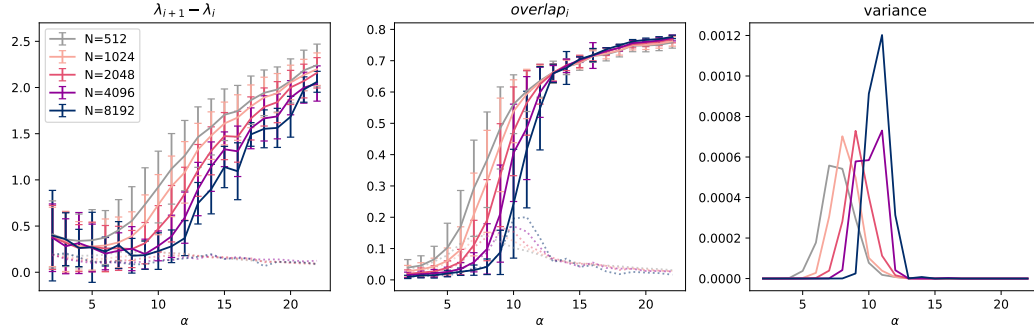

Figure 3: The three images show the occurring of a BBP transition at the moment when the training loss crosses the threshold energy. The images investigate the BBP as the input dimension is increased form $N = 512$ to $N = 8192$. At the BBP transition the smallest eigenvalue pops out of the bulk of eigenvalues and the associated eigenvector contains information on the signal. The left figure is the difference of the smallest eigenvalues (the second with the first in solid line, the third with the second in dotted line). The central image shows the overlap of the first eigenvector (full) and second eigenvector (dotted) with the signal. The transition appears to shift as in the left image. Finally on the right the fluctuations of the overlap first eigenvector with the teacher are shown, the peak corresponds to the transition.

Hessian is null. As the smallest eigenvalue of the Hessian is determined by the largest eigenvalue of $\mathcal{M}_{i,j}$, this imposes the additional condition $\lambda_1 = \lambda_2 = -\mu$. Using the second Eq. in (5) to enforce this last condition and the definition of $\mu$ in Eq. (3) one finds (see SM for more details):

$$\mu = \bar{\lambda}\mathbb{E}_{\hat{y},y}\left(\frac{\alpha_{\text{BBP}}\partial_{\hat{y}}^2\ell(\hat{y},y)y^2}{2\bar{\lambda}+\alpha_{\text{BBP}}\partial_{\hat{y}}^2\ell(\hat{y},y)}\right)\,, \qquad \mu = \frac{\alpha_{\text{BBP}}}{2}\mathbb{E}_{\hat{y},y}\left(\frac{\partial}{\partial\hat{y}}\ell(\hat{y},y)\hat{y}\right)\,. \qquad (7)$$

Together the three eqs. (6), and the two in (7) allow to determine the three unknown $\alpha_{\text{BBP}}, \bar{\lambda}, \mu$. Assuming that $\alpha_c = \alpha_{\text{BBP}}$, they therefore provide an equation for the algorithmic threshold once $P(\hat{y}, y)$ at threshold states is known.

## 2.2 Further numerical justifications

We now illustrate and test the theory by numerical simulations. Fig. 2 shows that the training loss slowly tends to the threshold energy, before departing in direction of the global minimum. According to the theory described in the previous section, the phenomenon occurring is a BBP transition. In order to confirm this we characterize the spectrum of the Hessian and focus on the smallest eigenvalues. On left figure of Fig. 2 we show the density of eigenvalues in blue and we show separately the smallest eigenvalue in orange and the second smallest eigenvalue in green. During the dynamics the spectrum moves compactly forming a bulk - except for the largest eigenvalues that do not play role in the transition. Approaching the threshold states, for $\alpha > \alpha_c$ the dynamics feels the presence of a descending direction associated with the smallest eigenvalue. This becomes increasingly strong and when the negative eigenvalue pops out of the bulk, the dynamics will follow and will converge to the global minimum. In the lower panel of Fig. 2 we show 5 snapshots of the spectral density during the dynamics.

In Fig. 3 we study the spectral properties of the Hessian at the time when the training loss hits the threshold energy. In the first panel from the left, we show the difference between the second smallest eigenvalue and the smallest eigenvalue (solid line) and the difference between the third smallest and the second smallest eigenvalue (dotted). The two quantities are close, and very small, until a certain value value of $\alpha$ - that depends on the size and is reported in the caption - after which the former increases linearly whereas the latter remains small. The second property that we investigate, central figure, is the overlap of the first and second eigenvectors with the teacher-vector $\boldsymbol{W}^*$ (respectively solid and dotted lines). This overlap is zero before the transition, then as $\lambda_2 - \lambda_1$, it suddenly increases and finally saturates. The first two panels are provide a strong evidences that a BBP transition is taking place. To further corroborate this findings, in the right panel we consider the fluctuation of the

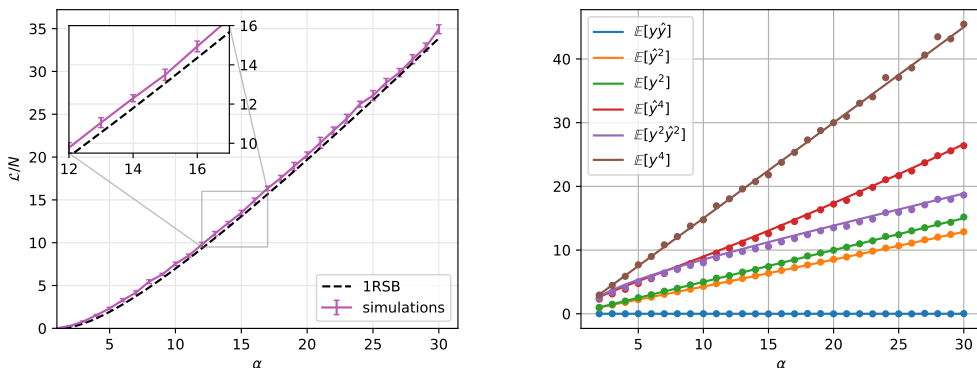

Figure 4: On the left panel is the loss of the threshold states from the simulations and from the analysis with the 1RSB replica method, evaluated for different values of $\alpha$. The errorbars given by mean and standard deviation of 1000 simulations with input dimension $N = 2048$ and shuffled labels. As expected increasing the number of datapoints the number of violated constraints increases and so does the threshold energies. On the right we compare the moments of the label distribution Eq. (17) (solid lines) with the moments obtained in the simulations (circles).

overlap (Fig. 3 right panel). In statistical physics terms, the overlap plays the role of order parameter, and its fluctuations diverge at the BBP transition. We indeed find that at the value of $\alpha$ at which the BBP transition seems to take place fluctuations peak (the more so the larger is $N$ as expected for a phase transition).

Finally, we also note the presence of strong finite size effects that shift the effective value of $\alpha$ at which the transition (cross-over for finite $N$) takes place.

## 3 Characterization of threshold states

The pivotal point of our analysis is that the gradient-flow dynamics is that the spurious minima trapping the dynamics and hindering weak recovery are the threshold states. The study of threshold states has been started in statistical physics of disordered systems, where it has been conjectured and verified that they play a prominent role in the gradient descent dynamics of such random systems [9]. according to this theory, threshold states are the most numerous minima and the Hessian associated to the threshold states has a spectrum that is positive semi-definite and gap-less.

In the previous section we have obtained a closed set of equations that allow us to obtain $\alpha_{\text{BBP}}$ once the distribution $\mathbb{P}(\hat{y}, y)$ is known. Such distribution can be derived using tools from statistical physics, in particular replica theory [10, 28, 29]. The main quantity of interest is the partition function, which is defined as the normalization constant of the Gibbs distribution associated with the loss Eq. (1),

$$\mathcal{Z} = \int_{\mathbb{S}^{\alpha(N-1)}(\sqrt{N})} e^{-\beta\mathcal{L}(\boldsymbol{W})} d\boldsymbol{W}, \tag{8}$$

where $\beta$ is a parameter associated to the inverse temperature in the physical analog of the problem. We consider the $\beta \to \infty$ limit to study the minima of $\mathcal{L}$ that are relevant in gradient-flow dynamics.

The disordered systems approach focuses on the average properties of the systems. In order to ensure concentration properties as the input dimension goes to infinity, the quantity to average is the logarithm of the partition function. This quantity in general is hard to compute and we resort to replica theory. The idea is to apply replica method, described below and in more detail in the SM, to move from the moments of the partition function to average of its logarithmic value. The analysis leads to the free energy density, Eq. (11), that we use to compute the distribution of the labels.

We start writing the moments of Eq. (8)

$$\overline{\mathcal{Z}^n} = \int_{\mathbb{S}^{\alpha(N-1)}(\sqrt{N})} \mathbb{E}_{(\boldsymbol{x},y)} e^{-\beta n\mathcal{L}(\boldsymbol{W})} d\boldsymbol{W}, \tag{9}$$

where we represent the average over the possible datasets with the overbar. The high-dimensional integral in Eq. (9) can be written in term of the overlap matrix $Q_{a,b} = \langle \boldsymbol{W}_a \cdot \boldsymbol{W}_b \rangle / N$ that encodes the similarity between two configurations extracted with the Gibbs measure, of the labels $\hat{y}_\mu$ and $y_\mu$, $\overline{\mathcal{Z}^n} = \int e^{NnS(\boldsymbol{Q})} \prod_{a \geq b=0}^n dQ_{a,b}$ with

$$
S(\boldsymbol{Q}) = \frac{1}{2} \log \det \boldsymbol{Q} + \alpha \log \exp \left[ \frac{1}{2} \sum_{a,b=0}^n Q_{a,b} \frac{\partial^2}{\partial h_a \partial h_b} \right] \exp \left[ -\frac{\beta}{2} \sum_a \ell(h_a, h_0) \right] \Bigg|_{\{h_a = 0 \; \forall a\}} . \tag{10}
$$

We perform a so-called *1-step replica symmetry breaking* (1RSB) analysis [30] that allows us to obtain an explicit expression for the distribution of $(\hat{y}, y)$. Using the 1RSB ansatz on this problem we reduce the number of parameters in Eq. (10) to $\chi$ and $z$: where $\chi$ is related to amplitude of the fluctuations in minimum when this is perturbed, and $z$ is a Lagrange multiplier that we use to enforce that the minima in consideration have a Hessian with gapless spectrum, i.e. they are threshold states. The general prescription for finding the global minimum of this kind of problem would require to optimize over $\chi$ and $z$, however, the same formalism can be used to characterize the threshold states by imposing a value for $z$ and optimizing only over $\chi$ [28, 31]. We evaluate the integral of the partition function Eq. (8) at the threshold using Laplace's method on Eq. (10). Finally we consider the replica method, by taking the analytic continuation of the moments of the partition function and the limit $n \to 0^+$. With this considerations we obtain the free energy density in low temperature limit $f(\chi|\alpha, z) = -\lim_{\beta \to \infty} \lim_{N \to \infty} \frac{1}{\beta N} \overline{\log \mathcal{Z}} / (\beta N)$ that reads

$$
f(\chi|\alpha, z) = -\frac{1}{2z} \log \frac{\chi + z}{\chi} - \frac{\alpha}{z} \int_{\mathbb{R}} dy \frac{1}{\sqrt{2\pi}} e^{-\frac{y^2}{2}} \log \gamma_1 \star e^{\frac{-z}{2} V(\hat{y}, y|\chi)} \Bigg|_{\hat{y}=0}, \tag{11}
$$

where $V(\hat{y}, y|\chi) \doteq \min_{\tilde{y}} \frac{(\hat{y} - \tilde{y})^2}{\chi} + \ell(\tilde{y}, y)$, the $\star$ symbol represent a convolution and we write $\gamma_1$ as a compact notation to represent a centered Gaussian distribution with unit variance. The arguments $\chi$ and $z$ are given via implicit equations that respectively impose that the Laplace's approximation on $f$ and that the free energy describes the threshold states (further details are in the SM):

$$
\frac{1}{\chi(\chi + z)} = \frac{\alpha}{4} \int_{-\infty}^\infty \frac{1}{\sqrt{2\pi}} e^{-\frac{y^2}{2}} \frac{\int_{-\infty}^\infty \frac{1}{\sqrt{2\pi}} e^{-\frac{\hat{y}^2}{2}} e^{-\frac{z}{2} V(\hat{y}, y|\chi)} \left( \partial_{\hat{y}} V(\hat{y}, y|\chi) \right)^2 d\hat{y}}{\int_{-\infty}^\infty \frac{1}{\sqrt{2\pi}} e^{-\frac{\hat{y}^2}{2}} e^{-\frac{z}{2} V(\hat{y}, y|\chi)} d\hat{y}} dy; \tag{12}
$$

$$
1 = \frac{\alpha}{4} \chi^2 \int_{-\infty}^\infty \frac{1}{\sqrt{2\pi}} e^{-\frac{y^2}{2}} \frac{\int_{-\infty}^\infty \frac{1}{\sqrt{2\pi}} e^{-\frac{\hat{y}^2}{2}} e^{-\frac{z}{2} V(\hat{y}, y|\chi)} \left( \partial_{\hat{y}}^2 V(\hat{y}, y|\chi) \right)^2 d\hat{y}}{\int_{-\infty}^\infty \frac{1}{\sqrt{2\pi}} e^{-\frac{\hat{y}^2}{2}} e^{-\frac{z}{2} V(\hat{y}, y|\chi)} d\hat{y}} dy. \tag{13}
$$

In order to obtain $\mathbb{P}(\hat{y}, y)$ we follow the strategy introduced in [29]. The partition function in Eq. (8) can be written as a functional integral over the empirical labels $\rho_{\text{quen}}(\hat{y}, y) = \frac{1}{\alpha N} \sum_{m=1}^{\alpha N} \delta(y - y_m) \delta(\hat{y} - \hat{y}_m)$:

$$
\mathcal{Z} = \int \mathcal{D}[\rho_{\text{quen}}(\hat{y}, y)] \, e^{-\frac{\beta \alpha N}{2} \int d\hat{y} dy \rho_{\text{quen}}(\hat{y}, y) \ell(\hat{y}, y) + N S[\rho_{\text{quen}}(\hat{y}, y)]}, \tag{14}
$$

where $S[\rho_{\text{quen}}(\hat{y}, y)]$ is the entropic factor (evaluated on configurations corresponding to the threshold states). This makes clear that the free-energy can also be obtained in terms of a large-deviation principle:

$$
f = \text{Min}_\rho \left( \frac{\alpha}{2} \int d\hat{y} dy \rho_{\text{quen}}(\hat{y}, y) \ell(\hat{y}, y) - S[\rho_{\text{quen}}(\hat{y}, y)] \right) \tag{15}
$$

The minimizer of this variational problem corresponds by definition to $\mathbb{P}(\hat{y}, y)$. By taking the functional derivative of $f$ with respect to $\ell(h, h_0)$ one obtains:

$$
\frac{\delta f}{\delta \ell(\hat{y}, y)} = \frac{\alpha}{2} \mathbb{P}(\hat{y}, y). \tag{16}
$$

Since we have an explicit expression of $f$ in terms of $\ell$, see eq. (11), we can obtain this distribution directly. In taking the functional derivative of Eq. (11) we must be careful in considering the implicit

dependence of $V(\hat{y}, y|\chi)$ from $\ell(\hat{y}, y)$. Finally inverting Eq. 9160 we obtain the label distribution

$$\mathbb{P}_{1\mathrm{RSB}}(\hat{y}, y) = \frac{1}{\sqrt{2\pi}} e^{-\frac{y^2}{2}} \frac{\exp\left[-\frac{\hat{y}^2}{2} - \frac{z}{2}V(\hat{y}, y|\chi)\right]}{\int_{\mathbb{R}} \exp\left[-\frac{\tilde{y}^2}{2} - \frac{z}{2}V(\tilde{y}, y|\chi)\right] d\tilde{y}}. \tag{17}$$

The index 1RSB denotes that the free-energy has been obtained within the 1RSB scheme.

We run numerical experiments to characterize the threshold states in terms of their energy and their moments with respect to the label probability distribution. In Fig. 4 (left) the 1RSB threshold energy is plotted together with the value of the plateau obtained from the simulations in Fig. 1. The 1RSB ansatz appears to be a good approximation of the empirically obtained energy. In the inset we highlight the discrepancy with the empirical line, which may be due to both finite size effects as well as to the fact that the 1RSB scheme is an approximation and more involved schemes must be employed to obtain the exact distribution [30]. Finally in Fig. 4 (right) we show the first 4 moments of the label distribution from the 1RSB analysis and we compare them with the numerical results. The 1RSB moments give a nice agreement in relation to the empirical ones.

Encouraged by the reasonable, though not perfect, accuracy of the 1RSB approximation, we use the expression (17) as input for the three eqs. in (6,7). Their solutions leads to a finite threshold at $\alpha_{\mathrm{BBP}}^{\mathrm{1RSB}} \approx 13.8$. This value could be compatible with the numerical results presented in Fig. 3 if the finite size shift saturates for yet larger sizes.

## 4   Discussion

In Fig. 5 we present the fraction of runs of gradient descent that converge to zero test error in the phase retrieval problem under consideration. We see that a large fraction of simulations achieves convergence before the $\alpha_{\mathrm{BBP}}^{\mathrm{1RSB}}$ threshold, in general before $\alpha \approx 11$. The mechanism behind this difference is not clear to us, as well as it is not clear whether it will hold as the input dimension tends to infinity. The numerical results are affected by strong finite size effects and may be both consistent with a transition taking place at finite alpha in the infinite dimensional limit and with curves that shift in alpha as a logarithm in the input dimension. Empirically we do not observe any unsuccess above $\alpha_{\mathrm{BBP}}^{\mathrm{1RSB}}$ which is consistent with our theoretical scenario for any $N > 64$. In [7] the authors showed that large finite size effects in the initialization affect the location of the BBP transition. Whether this happens also in this case is unclear and deserves further investigations.

Let us conclude by commenting on the limitations and possible generalizations of the results presented in this paper. A key element of the phase retrieval model is the existence of a phase at small $\alpha$ in which the best achievable generalization error is not better than random guess. This arises in models that present a symmetry, such as the $\pm 1$ symmetry due to the absolute value in the phase retrieval problem. The picture shown in Fig. 1 where the threshold states are characterized using the gradient-flow dynamics in a variant of the model with randomized labels is enabled by this symmetry and the rest of our analysis relies on the simplifications stemming from this. We expect that the BBP picture presented in this work generalizes to all learning where at small sample complexity error is not better than random guess. Of course working out the details, even for two layer neural networks, is an open problem of interest for future work. In most learning problems a naive linear regression often achieves a better than random guess performance, and thus an extension of our theory for problems of that type (lacking a symmetry) would be an interesting direction for future work.

## Acknowledgments and Disclosure of Funding

We thank Carlo Lucibello for precious discussions and Federico Ricci-Tersenghi for interesting comments on the manuscript. We acknowledge funding from the ERC under theEuropean Union's Horizon 2020 Research and Innovation Programme Grant Agreement 714608-SMiLe;from the French government under management of Agence Nationale de la Recherche (ANR) grant PAIL,and grant "Investissements d'Avenir" LabEx PALM (ANR-10-LABX-0039-PALM) (StatPhysDisSys) and(ANR-19-P3IA-0001) (PRAIRIE 3IA Institute); and from the Simons Foundation collaboration Crackingthe Glass Problem (#454935, Giulio Biroli).

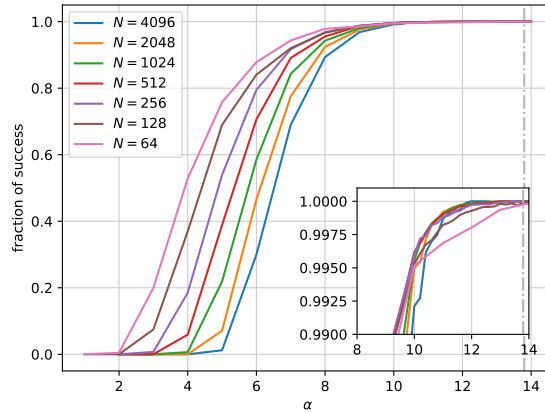

Figure 5: Fraction of simulations with gradient descent that achieve zero test error for various input dimensions $N$. The number of simulations increases with the smaller input dimension in order the account for the larger fluctuations. The number varies from $10000$ simulations for $N = 4192$ to $100000$ simulations for $N = 64$. The learning rate is $0.0002$ and the simulation runs until either the loss goes below $10^{-8}$ or the number of steps exceeds $1500 \log_2 N$ iterations. The logarithmic term in the time accounts for the fact that the average initial overlap with the ground truth is $\sim 1/\sqrt{N}$. In the inset we zoom in the region where all the simulations converge showing the indication of a crossing with large fluctuations due to finite size effects.

## Broader Impact

Our work is theoretical in nature, and as such the potential societal consequence are difficult to foresee. We anticipate that deeper theoretical understanding of the functioning of machine learning systems will lead to their improvement in the long term.

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
