[Supplementary Material]

# Supplementary Material of
# Complex Dynamics in Simple Neural Networks: Understanding Gradient Flow in Phase Retrieval

**Stefano Sarao Mannelli**[1], **Giulio Biroli**[2], **Chiara Cammarota**[3,4],
**Florent Krzakala**[2], **Pierfrancesco Urbani**[1], **and Lenka Zdeborová**[1]

## A    BBP transition and phase transition of spectra initialization

In non-convex estimation problems, such as Phase Retrieval, big advantages follow from the development of well tailored spectral methods to be used as initialization step. Recently the outcomes of one such spectral method widely used in Phase Retrieval and based on the construction of a data matrix

$$\mathcal{M}_{i,j} = \frac{1}{\alpha N} \sum_{m=1}^{\alpha N} \mathcal{T}(Y_m) u_{m,i} u_{m,j} = \frac{1}{\alpha} \sum_{m=1}^{\alpha N} \mathcal{T}(Y_m) x_{m,i} x_{m,j} \tag{A.1}$$

from sensing vectors $\boldsymbol{u}_m$, with elements of order one (or sensing vectors $\boldsymbol{x}_m$ on the unitary sphere, according to our definition) and measurements $Y_m$ has been exactly derived [1]. The method involves a pre processing function $\mathcal{T}(Y_m)$ that can be optimized to further improve the results. Once the data matrix is constructed the eigenvector $\boldsymbol{v_1}$ corresponding to the largest eigenvalue $\lambda_1$ can be used as an estimator of the signal $\boldsymbol{W}^*$.

To obtain the performances of this kind of spectral initialization it is assumed [1] that the measurements $Y_m$ are independently drawn according to a density function conditional on $y_m = \langle \boldsymbol{x}_m \boldsymbol{W}^* \rangle$ associated to the particular acquisition process, and it is recalled that $y_m$ are themselves Gaussian random variables, due to the definition of the problem. Finally in the large $N$ limit the empirical average used to construct the data matrix can be replaced by the expected value $\mathbb{E}_{Y,y}$ over these two distributions.

The result [1] goes as follows. Given two functions defined as

$$\Psi_\alpha(\lambda) \equiv \lambda \left[ \frac{1}{\alpha} + \mathbb{E}_{Y,y} \left( \frac{\mathcal{T}(Y)}{\lambda - \mathcal{T}(Y)} \right) \right] \tag{A.2}$$

and

$$\Phi(\lambda) = \lambda \mathbb{E}_{Y,y} \left( \frac{\mathcal{T}(Y) y^2}{\lambda - \mathcal{T}(Y)} \right) \tag{A.3}$$

with $\lambda > \mathcal{T}(Y)$, and given

$$\xi_\alpha(\lambda) = \Psi_\alpha(\max\{\lambda, \bar{\lambda}\}) \tag{A.4}$$

with

$$\bar{\lambda} = \arg\min \Psi_\alpha(\lambda) \,, \tag{A.5}$$

the two largest eigenvalues of $\mathcal{M}$, $\lambda_1$ and $\lambda_2$, are such that

$$\lambda_1 \to_{\mathbb{P}} \xi_\alpha(\lambda_\alpha^*) \,, \tag{A.6}$$

1. Université Paris-Saclay, CNRS, CEA, Institut de physique théorique, 91191, Gif-sur-Yvette, France.
2. Laboratoire de Physique de l'Ecole normale supérieure ENS, Université PSL, CNRS, Sorbonne Université, Université Paris-Diderot, Sorbonne Paris Cité Paris, France
3. Dipartimento di Fisica, Sapienza Università di Roma, P.le A. Moro 5, 00185 Rome, Italy
4. Department of Mathematics, King's College London, Strand London WC2R 2LS, UK

with $\lambda_\alpha^*$ the solution of $\xi_\alpha(\lambda) = \Phi(\lambda)$, and

$$\lambda_2 \to_\mathbb{P} \Psi_\alpha(\bar{\lambda}). \tag{A.7}$$

A phase transition occurs at the largest $\alpha$ such that $\lambda_1 = \lambda_2$, which can be evaluated by imposing $\Psi_\alpha'(\lambda_\alpha^*) = 0$ or equivalently

$$\frac{1}{\alpha} = \mathbb{E}_{Y,y}\left(\frac{\mathcal{T}(Y)^2}{(\bar{\lambda} - \mathcal{T}(Y))^2}\right) , \tag{A.8}$$

which corresponds to $\Psi_\alpha'(\bar{\lambda}) = 0$ as at that point $\lambda_\alpha^* = \bar{\lambda}$. At larger $\alpha$ the largest eigenvalue pops out from the spectrum bulk ($\lambda_1 \neq \lambda_2$) and the corresponding eigenvector develops a finite correlation with the signal, in a phenomenon called BBP transition [2], hence the definition of $\alpha_{\mathrm{BBP}}$ when this occurs.

It is interesting to note that the structure of the data matrix is closely reminiscent of the structure of the first term in the Hessian of our problem (see Eq. (4)). Indeed incidentally the original idea of this spectral method for initialization can be traced back to the study of Hessian's principal directions [3]. In particular we observe that

$$-\mathcal{H}_{i,j}(\boldsymbol{W}) = -\frac{1}{2}\sum_{m=1}^{\alpha N}\frac{\partial^2}{\partial \hat{y}_m^2}\ell(\hat{y}_m, y_m)x_{m,i}x_{m,j} + \mu\delta_{i,j} = \mathcal{M}_{i,j} + \mu\delta_{i,j} , \tag{A.9}$$

provided that the pre processing function is

$$\mathcal{T}(Y) = -\frac{\alpha}{2}\frac{\partial^2}{\partial \hat{y}^2}\ell(\hat{y}, y) = -\frac{\alpha}{2}\partial_{\hat{y}}^2\ell(\hat{y}, y) . \tag{A.10}$$

Note that we consider a case for which measurements $Y$ have a one to one correspondence with $y$ (*i.e.* $Y = y^2$). Moreover in the problem discussed empirical averages involve not only measurements $Y$ but also estimated $\hat{y} = \langle \boldsymbol{xW}\rangle$ in correspondence of some $\boldsymbol{W}$ of interest. Therefore the expected value, $\mathbb{E}_{\hat{y},y}$, should be taken over the relative joint probability distribution function $P(\hat{y}, y)$. In conclusion, the results just mentioned tell immediately what is the largest $\alpha$ (*i.e.* $\alpha_{BBP}$):

$$\frac{1}{\alpha_{\mathrm{BBP}}} = \mathbb{E}_{\hat{y},y}\left(\frac{\alpha_{\mathrm{BBP}}\partial_{\hat{y}}^2\ell(\hat{y}, y)}{2\bar{\lambda} + \alpha_{\mathrm{BBP}}\partial_{\hat{y}}^2\ell(\hat{y}, y)}\right)^2 , \tag{A.11}$$

before the smallest eigenvalue of the Hessian pops out from the spectrum bulk, being also associated to an eigenvector with finite projection on the signal to be detected.

In this case we are focusing on performance of a gradient descent dynamics, which for mean-field spin-glasses naturally gets stuck on what are called threshold states [4]. We argue that the gradient descent dynamics applied to Phase Retrieval, when retrieval fails, will also approach threshold states which are mainly characterized by their property of being marginal, *i.e.* the smallest eigenvalue of their Hessian is null. This qualifies the relevant $\boldsymbol{W}$ as a typical configuration belonging to threshold states and $P(\hat{y}, y)$ as the joint probability distribution at threshold states. Moreover it allows to introduce the marginal condition $\lambda_2 = \lambda_1 = -\mu$, which can be re-expressed by equating $\lambda_2 = \Psi_\alpha(\bar{\lambda})$ to $-\mu$:

$$\mu = \bar{\lambda}\mathbb{E}_{\hat{y},y}\left(\frac{\alpha_{\mathrm{BBP}}\partial_{\hat{y}}^2\ell(\hat{y}, y)y^2}{2\bar{\lambda} + \alpha_{\mathrm{BBP}}\partial_{\hat{y}}^2\ell(\hat{y}, y)}\right) . \tag{A.12}$$

Finally the definition of the spherical parameter (3) in the main text must be considered to close the system of equations

$$\mu = \frac{\alpha_{\mathrm{BBP}}}{2}\mathbb{E}_{\hat{y},y}\left(\partial_{\hat{y}}\ell(\hat{y}, y)\hat{y}\right) \tag{A.13}$$

to be used to determine $\alpha_{\mathrm{BBP}}, \bar{\lambda}, \mu$ in correspondence of $P(\hat{y}, y)$ from threshold states.

The resulting picture is as follows. In the small $\alpha$ regime, gradient descent will systematically approach threshold states and remain stuck there. However starting from $\alpha_{\mathrm{BBP}}$ in the Hessian of these states, that is otherwise marginal, an isolated eigenvalue pops out from the bulk immediately becoming negative. Moreover the eigenvector associated to such negative direction, naturally followed by gradient descent, has a finite overlap with the signal. Therefore, we argue, it is from that point on that the signal should be easily retrieved.

# B  Replica Analysis

The field of physics of disordered systems has developed numerous tools to deal with random systems [5, 6]. At an abstract level of thinking, using those tools means that we identify the inference problem with a physical systems that subject to a certain potential. The randomness comes from the having a dataset made of random projections. The estimator is mapped into a spherical spin and the loss function becomes the energy - or *Hamiltonian*. Finally the system the temperature in which the system lives is sent to zero and the system tend to the lowest energy, herefore minimizing the loss. The ground truth in the inference problem becomes equivalent to a minimum planted in the energetic landscape [7]. For example this formulation is equivalent to a physical system that before the experiment is liquid, but as we cool it down it can become either a crystal - an ordered solid - or a glass - an amorphous solid. Finding the crystal means reconstructing the signal.

## B.1  Partition function

Moving to the mathematical aspects of the problem. We define a Gibbs distribution associated with the problem and evaluate its normalization constant - the *partition function* $\mathcal{Z}$. The partition function, and in particular its logarithm divided by the temperature - the free energy -, contains all the information we aim to understand in the problem. By taking the proper derivatives, and possibly add an external field, we can compute relevant macroscopic properties such as: average overlap with the ground truth, average loss achieved. In disordered system we have to consider the additional complication given by the randomness. Therefor, we need to consider the average free energy that is the average of the logarithm of a high-dimensional integral, that can be done by the simple observation:

$$\log \mathcal{Z} = \lim_{n \to 0} \frac{\mathcal{Z}^n - 1}{n}. \tag{B.1}$$

Which for arbitrary $n$ is not simpler than computing the logarithm, but it is much simpler if $n \in \mathbb{N}$ and we perform an analytic continuation to $n \in \mathbb{R}$. Under this - replica trick - the average of the logarithm is equivalent to compute the average of the $n$th moment of the partition function and take the limit. Formally the $n$th moment correspond to the partition function of $n$ identical - replicated - system that do not interact but with the same realization of the disorder.

The problem is now computing the moments of the partition function which in general is prohibitive and we have to use an ansatz on the specific form of the solution, in particular we use the so called *replica symmetry breaking ansatz* [5]. This largely reduces the number of parameters. Finally the average free energy can be evaluated by set of saddle point equations.

We can now move to the analysis. The partition function already defined in the main text is

$$\mathcal{Z} = \int_{\mathbb{S}^{N-1}(\sqrt{N})} \mathcal{D}\boldsymbol{W}_S \, \exp\left\{ -\frac{\beta}{2} \sum_{\mu} \ell(\langle \boldsymbol{x}_{\mu}, \boldsymbol{W}_S \rangle, \langle \boldsymbol{x}_{\mu}, \boldsymbol{W}_T \rangle) \right\} \tag{B.2}$$

and consider its $n$th moment - the *replicated partition function* -

$$\overline{\mathcal{Z}^n} = \mathbb{E}_{\{\boldsymbol{x}_{\mu}\}} \int_{\mathbb{S}^{n(N-1)}(\sqrt{N})} \prod_{a=1}^{n} \mathcal{D}\boldsymbol{W}_a \, \exp\left\{ -\frac{\beta}{2} \sum_{\mu} \ell(|\langle \boldsymbol{x}_{\mu}, \boldsymbol{W}_S \rangle|, |\langle \boldsymbol{x}_{\mu}, \boldsymbol{W}_T \rangle|) \right\}. \tag{B.3}$$

This is formally equivalent to have $n$ independent systems. Introduce the overlaps with the projector $r_{\mu}^{(a)} = \langle \boldsymbol{x}_{\mu}, \boldsymbol{W}_a \rangle$ with indices $a = 0, \ldots, n$ where $a = 0$ is the overlap with the ground truth and the others are the overlaps with the estimators of the $n$ systems. Introduce those quantities in the replicated partition function via Dirac's deltas using their Fourier transform

$$\overline{\mathcal{Z}^n} = \mathbb{E}_{\{\boldsymbol{x}_{\mu}\}} \int \prod_a \mathcal{D}\boldsymbol{W}_a \int \mathcal{D}(r, \hat{r}) \exp\left\{ -\frac{\beta}{2} \sum_{a,\mu} \ell(r_{\mu}^{(a)}, r_{\mu}^{(0)}) + i \sum_{a,\mu} \hat{r}_{\mu}^{(a)} r_{\mu}^{(a)} - i \sum_{a,\mu} \langle \boldsymbol{x}_{\mu}, \hat{r}_{\mu}^{(a)} \boldsymbol{W}_a \rangle \right\} =$$

$$= \int \prod_a \mathcal{D}\boldsymbol{W}_a \int \mathcal{D}(r, \hat{r}) \exp\left\{ -\frac{\beta}{2} \sum_{a,\mu} \ell(r_{\mu}^{(a)}, r_{\mu}^{(0)}) + i \sum_{a,\mu} \hat{r}_{\mu}^{(a)} r_{\mu}^{(a)} - \frac{1}{2N} \sum_{\mu} \sum_{a,b=0}^{n} \hat{r}_{\mu}^{(a)} \hat{r}_{\mu}^{(b)} \langle \boldsymbol{W}_a, \boldsymbol{W}_b \rangle \right\}. \tag{B.4}$$

Where $\mathcal{D}$ contains the normalization factor of the Fourier transform. We introduce the matrix of overlaps between estimators and ground ground truth $\boldsymbol{Q}$, $Q_{ab} = \frac{1}{N}\langle \boldsymbol{W}_a, \boldsymbol{W}_b \rangle$. This is done using the same idea of introducing delta function and gives a contribution $\frac{N}{2} \log \det \boldsymbol{Q}$ to the action [8,9]. The equation can now be factorized in $\mu$, so we can drop the $\mu$s from $r$ and $\hat{r}$. Observing that

$$e^{-\frac{1}{2}\sum_{ab} \hat{r}^{(a)}\hat{r}^{(b)} Q_{ab}} = e^{\frac{1}{2}\sum_{ab} Q_{ab} \frac{\partial^2}{\partial h_a \partial h_b}} e^{-i\, h_a \hat{r}^{(a)}}\Big|_{\{h_a=0\}_a}, \tag{B.5}$$

we can integrate over $r$ and $\hat{r}$, and write a simplified replicated partition function $\overline{\mathcal{Z}^n} = \int \prod_{a \geq b=0}^{n} dQ_{ab} e^{NS(\boldsymbol{Q})}$ with action

$$S(\boldsymbol{Q}) = \frac{1}{2}\log\det\boldsymbol{Q} + \alpha\log\exp\left[\frac{1}{2}\sum_{a,b=0}^{n} Q_{a,b}\frac{\partial^2}{\partial h_a \partial h_b}\right]\exp\left[-\frac{\beta}{2}\sum_a \ell(h_0, h_a)\right]\Big|_{\{h_a=0\}_a}. \tag{B.6}$$

where the first term is an entropic term that accounts for the degeneracy of the matrix $\boldsymbol{Q}$ in the space of symmetric matrices. And the second term is an energetic term that accounts for the potential acting on the system.

Observe that so far we did not make any ansatz on the structure of the overlap matrix $\boldsymbol{Q}$. In the next subsections we will consider the 1-step replica symmetry breaking ansatz (1RSB).

## B.2   1 step replica symmetry breaking

The 1RSB scheme consists in making an ansatz on the structure of the overlap matrix $\boldsymbol{Q}$ [5]. The assumption is that not all the replicated systems will have the same overlap - which correspond to the replica symmetric ansatz - but the systems are clustered. Systems inside the same cluster will have a larger overlap, systems outside the cluster will have a smaller overlap. This translates into the following parameters: $q_1$ overlaps inside the same cluster, $q_0$ overlaps in different clusters, $x$ dimension of the clusters, finally $m$ the overlap with the signal. Schematically we have

$$\boldsymbol{Q} = \begin{pmatrix} 1 & & m & & & m & \\ & \begin{pmatrix} 1 & q_1 & q_1 \\ q_1 & 1 & q_1 \\ q_1 & q_1 & 1 \end{pmatrix} & & & q_0 & \\ m & & & & \begin{pmatrix} 1 & q_1 & q_1 \\ q_1 & 1 & q_1 \\ q_1 & q_1 & 1 \end{pmatrix} & \\ m & & q_0 & & & & \end{pmatrix} \tag{B.7}$$

with $\boldsymbol{Q}$ of dimension $(n+1) \times (n+1)$, and the inner matrices of dimension $x \times x$.

The analysis proceed as in a standard way: we derive the 1RSB free energy with the associated saddle point equations and move to the zero temperature limit [5,6], we impose the marginal stability - corresponding to the threshold states - [10], finally we derive the label distribution [11].

For notational convenience we call $\gamma_a(x)$ the probability density function of a Gaussian with zero mean and (co-)variance $a$, and we use the symbol $\star$ to indicate the convolutions, i.e. $f \star g(x) = \int_{\mathbb{R}} f(x-t)g(t)dt$.

We can plug Eq. (B.7) into Eq. (B.6) and obtain 1RSB formulation of the action.

$$\frac{1}{n}S_{1\mathrm{RSB}}(\boldsymbol{Q}) = \frac{1}{2}\log(1-q_1) + \frac{1}{2x}\log\frac{1-q_1+x(q_1-q_0)}{1-q_1} + \frac{1}{2}\frac{q_0-m^2}{1-q_1+x(q_1-q_0)} + $$
$$+ \alpha\int_{\mathbb{R}^2} d\hat{y}dy\gamma_{\boldsymbol{\Sigma}}(\hat{y},y)\frac{1}{x}\log\left[\gamma_{q_1-q_0}\star\left(\gamma_{1-q_1}\star e^{-\frac{\beta}{2}\ell(\hat{y},y)}\right)^x\right]. \tag{B.8}$$

and using Eq. (B.1) the free energy is $-\frac{1}{n\beta}S_{1\mathrm{RSB}}$.

We can now take derivatives to find the saddle point equations.

$$\frac{1}{x}\left(\frac{1}{1-q_1}-\frac{1}{1-q_1+x\,(q_0-q_1)}\right)+\frac{q_0-m^2}{[1-q_1+x\,(q_0-q_1)]^2}=$$
$$=\alpha\int_{\mathbb{R}^2}d\hat{y}dy\gamma_{\Sigma}(\hat{y},y)e^{-x\,f(x,\hat{y},y)}\gamma_{q_1-q_0}\star\left[e^{x\,f(1,\hat{y},y)}\left(\frac{d}{d\hat{y}}f(1,\hat{y},y)\right)^2\right]; \tag{B.9}$$

$$\frac{q_0-m^2}{[1-q_1+x\,(q_0-q_1)]^2}=\alpha\int_{\mathbb{R}^2}d\hat{y}dy\gamma_{\Sigma}(\hat{y},y)\left(\frac{d}{dh}f(x,\hat{y},y)\right)^2; \tag{B.10}$$

$$\frac{m}{1-q_1+x\,(q_0-q_1)}=\alpha\int_{\mathbb{R}^2}d\hat{y}dy\gamma_{\Sigma}(\hat{y},y)\frac{d^2}{d\hat{y}dy}f(x,\hat{y},\hat{y}); \tag{B.11}$$

where we define

$$f(1,\hat{y},y)=\log\left[\gamma_{1-q_1}\star e^{-\frac{\beta}{2}\ell(\hat{y},y)}\right];$$
$$f(x,\hat{y},y)=\frac{1}{x}\log\left[\gamma_{q_1-q_0}\star\left(\gamma_{1-q_1}\star e^{-\frac{\beta}{2}\ell(\hat{y},y)}\right)^x\right]=\frac{1}{x}\log\left[\gamma_{q_1-q_0}\star e^{xf(1,\hat{y},y)}\right];$$
$$\ell(\hat{y},y)=(\hat{y}^2-y^2)^2;$$

and $\Sigma$ is a $2\times 2$ covariance matrix with entries $\Sigma_{11}=1$, $\Sigma_{12}=m$ and $\Sigma_{22}=q_0$.

**Zero temperature and parameter ansatz**   As the zero temperature goes to zero the order parameters $q_1$ and $x$ needs to be rescaled with the temperature as $q_1\approx 1-\chi T$ and $x\approx zT$ with $\chi,z\sim O(1)$. Instead $m$ and $q_0$ will not be affected by the limit. The reason why some parameters need to be rescaled is that as long as there is a positive temperature the replicated systems exploit this thermal energy to fluctuate in the basin of the minimum, therefor their overlap $q_1$ is given by average overlap in the basin of attraction. When the temperature drops to zero the thermal goes to zero and all the system shrink to a point. $\chi$ represent the fluctuation that systems can have when they receive an infinitesimal amount of thermal energy. With the same physical reason, also the cluster itself shrinks to a point. The rate of convergence to the point is given by $z$.

Under those observation we obtain the 1RSB free energy

$$-f(\boldsymbol{Q})\approx_{\beta\gg 1}\frac{1}{2z}\log\frac{\chi+z(1-q_0)}{\chi}+\frac{1}{2}\frac{q_0}{\chi+z(1-q_0)}+$$
$$+\frac{\alpha}{y}\int_{\mathbb{R}}dy\gamma_{\boldsymbol{\Sigma}}(\hat{y},y)\log\gamma_{1-q_0}\star e^{\frac{-y}{2}V(\hat{y},y|\chi)}\Bigg|_{\hat{y}=0}. \tag{B.12}$$

and the corresponding saddle point equations from Eqs. (B.9-B.11).

However, the solution of those equation will lead to the global minimum of the loss, while we are interested in the threshold states. We follow the idea of [10] where $z$ is used as a Lagrange multiplier that selects the threshold states. $z$ is fixed by imposing the marginal stability condition, i.e. that the spectrum of the Hessian of the minima in consideration touches zero, following [11] this is given by

$$1=\frac{\alpha}{4}\chi^2\int_{\mathbb{R}}dy\frac{1}{\sqrt{2\pi}}e^{-\frac{y^2}{2}}\frac{\gamma_1\star\left[V''(\hat{y},y|\chi)^2e^{-\frac{z}{2}V(\hat{y},y|\chi)}\right]}{\gamma_1\star e^{-\frac{z}{2}V(\hat{y},y|\chi)}}\Bigg|_{\hat{y}=0}. \tag{B.13}$$

At the threshold the overlap with the signal is zero, $m=0$, and for symmetries of the problem also $q_0=0$. In fact $m=0$ is always a solution of Eq. (B.11), and if $m=0$ then $q_0=0$ is also solution of the second saddle point equation, Eq. (B.10), as in the RHS the Gaussian becomes degenerate in $h$ and $d_hf(x,h,h_0)|_{h=0}$ being an odd function in $h$. Therefore we can restrict the equations to just the one for $\chi$, Eq. (B.9) that becomes

$$\frac{\beta^2}{z}\left(\frac{1}{\chi}-\frac{1}{\chi+z}\right)=\alpha\int_{\mathbb{R}}dy\gamma_1(y)e^{-x\,f(x,0,y)}\int_{\mathbb{R}}d\hat{y}\gamma_1(\hat{y})\left[e^{x\,f(1,\hat{y},y)}\left(\frac{d}{d\hat{y}}f(1,\hat{y},y)\right)^2\right]. \tag{B.14}$$

Rewriting these equation expliciting the convolutions we obtain the equations presented in the main text. With those element we can obtain the label distribution presented in the main text.

# C Additional details on the numerical experiments

## C.1 Dynamics with shuffled labels

In the main text we showed that the dynamics is always attracted by the threshold states before (possibly) finding the good direction that leads to the optimal solution. To visually complement the explanation, in Fig. 1 we consider the dynamics in the same setting of Fig. 2, same dataset and same initialization, but using shuffled labels. The upper figure shows the density of eigenvalues during the evolution while the lower figure considers the histogram for five cuts at iteration 1, 453, 906, 1208 and 1510. We remark that in the initial stages of their evolution the two simulations follow the same dynamics (iteration 1, 453, and 906). Only after a transient the simulation with the correct labels shows the presence of the BBP transition (iteration 1208) and converge to the solution (iteration 1510). Instead, the simulation with the shuffled labels keeps approaching the threshold states for the rest of the simulation time tending to the marginal condition. We remark that in the upper figure the smallest (dashed orange line) and second smallest eigenvalues (solid green line) that are always very close, as it is expect since in this case the distribution of the eigenvalues always forms a bulk.

Figure 1: Comparison of the properties of the Hessian for phase retrieval between the problem with the correct labels and the problem with shuffled labels. The system simulated has size $N = 2048$ and $\alpha = 10$. The database considered in the simulation in the upper figure is the one used for Fig. 2 with the shuffled labels. The figures represent the same quantities plotted in Fig. 2 and are intended to make the comparison between the evolution in the two settings. The figure below represents the distribution of the eigenvalues during the dynamics for the problem with correct and shuffled labels, respectively in blue and orange.

## C.2 Numerical threshold

**On the choice of the learning rate.** The dynamics considered in the main text is gradient flow, while numerically we rely on the discretized version of the algorithm, namely gradient descent. In order to have agreement between the theoretical analysis and the simulations we must consider a learning rate sufficiently small to reduce the discrepancy between the two algorithms. We consider different input dimensions and tested different learning rates, in Fig. 2 we report this process for $N = 1024$. In the figure on the left we can observe that the dynamics of 10 simulations at $\alpha = 7$ testing the learning rates : $\eta = 2 \times 10^{-4}$ (solid lines), $\eta = 1 \times 10^{-4}$ (dashed lines), $\eta = 5 \times 10^{-5}$ (dotted lines). We can notice that the lines are nicely overlapping, this give us the learning rate $\eta = 2 \times 10^{-4}$ that we use in our simulations. In the right figure we show the fraction of successful simulations, where success is defined as the fact that the overlap $\boldsymbol{W} \cdot \boldsymbol{W}^*/N > 0.99$. Notice that the line is shifting until we reach the learning rate $\eta = 2 \times 10^{-4}$.

Figure 2: Choice of the learning rate for the phase retrieval problem with input dimension $N = 1024$. On the left we show the overlap $\boldsymbol{W} \cdot \boldsymbol{W}^*/N$ in time for 10 different simulations at $\alpha = 7$. The different line styles refer to different learning rates. On the right we show the fraction of success, as reported in Fig. 5, as we change the learning rate. In the simulations we adapted the stopping criterion to the learning rate so that the curve get to the same simulation time, more specifically for the left panel the stopping times are : $1000 \log_2 N$ for $\eta = 2 \times 10^{-4}$, $2000 \log_2 N$ for $\eta = 1 \times 10^{-4}$, and $4000 \log_2 N$ for $\eta = 5 \times 10^{-5}$.

**On the nature of the transition.** Our analysis assumes a first order transition between: the easy phase, where gradient flow with high probability solves the problem; and the hard phase, where gradient flow with high probability does not find the optimal solution and this solution is information theoretically achievable. According to this hypothesis threshold identifies a jump in the overlap between estimator and ground truth, which defines the first order transition. We bring as additional support on the matter Fig. 3, where we show the final overlaps reached by the simulations for different size of the dataset $\alpha$. As $\alpha$ increases, the figures show that densities become bimodal and finally concentrate on 1 meaning that the problem is easy. This is an indicator of a first order transition. Contrary to the second order (continuous) transition, where the distribution of the overlaps should be unimodal.

Figure 3: The five figures show the final overlaps between estimator and teacher for a large number of simulations at different values of $\alpha$. The figure refer to different input dimensions, from left to right : 256, 512, 1024, 2048, and 4096.