[Reviews · NeurIPS 2020]

Review 1

Summary and Contributions: The authors study phase retrieval with Gaussian data with gradient flow, in an asymptotic regime of infinite input size, number of samples and training time. The authors present evidence that gradient flow can get stuck in so called threshold states which are spurious local minimizers if not enough samples are used. At a certain critical value of the sample complexity relative to the dimension alpha, a phase transition in the structure of the Hessian at these points appears to occur and a direction of negative curvature emerges. Using tools from statistical physics of disordered systems and under some additional assumptions about the structure of the threshold states, their distribution and the critical value of alpha are computed. The authors also present empirical evidence illustrating the phenomena discussed. The critical value of alpha computed seems to differ from the values observed in experiment, possibly because of finite system size effects. Also, the values of alpha at which recovery is possible with reasonable probability appears much lower than the computed value. I thank the authors for the explanations in the rebuttal and the additional figures provided. I still believe the paper should be accepted.

Strengths: Problems in machine learning that involve solving a nonconvex optimization problem with gradient descent are ubiquitous and still somewhat poorly understood theoretically, and understanding this phenomenon is an important goal for the field. One way to better understand it is the detailed study of nonconvex optimization problems that can be solved efficiently with gradient descent, and this work studied one such problem. The empirical evidence of the phenomena described seem compelling. There also appears to be good agreement between the leading moments of the distribution of the threshold states as evaluated empirically and the computed moments.

Weaknesses: One of the main assumptions used in characterizing the "threshold states" at which the gradient flow dynamics appear to get trapped is that the Hessian is positive semidefinite. On the other hand, in figure 2 as the training loss crosses the threshold energy the minimal eigenvalues of the Hessian appear clearly negative, unless I am misunderstanding the figure. The authors do not appear to address this point. Could the soundness of this assumption also account for the inaccuracy of the computed value of alpha at which the phase transition occurs which can be seen in figure 4? The main result of the paper - the computation of the relative sample size alpha at which the phase transition occurs, doesn't seem to be very accurate when compared to experiments in figure 3 and 5. It would have also been helpful to plot this value in the figure in order to make this point clear. The discrepancy could be a result of finite size effects as the authors claim, but could also be a result of say the assumption made about the Hessian at the threshold states or the accuracy of the 1RSB ansatz. The fact that recovery is possible for much lower values of alpha than those predicted by the calculation, as seen in figure 5, raises a suspicion that there may be another mechanism in play, at least in the setting in which the experiment was conducted. Insufficient details are provided about the experiments performed. In particular it is unclear how the learning rate chosen affects the dynamics.

Correctness: The methodology is a widely accepted one in statistical physics, including a number of assumptions made and steps in the derivation that are not rigorous yet yield accurate predictions in many known cases.

Clarity: The paper is well written overall. A number of results from previous works are quoted and used. The clarity could be improved if these are introduced in greater detail. While this is impractical to do in the main text due to space constraints, a more detailed overview of these in the supplementary materials would have been helpful. The critical value of alpha from the calculation could have been added to figure 3 for clarity.

Relation to Prior Work: Yes, the details of prior work as well as key distinctions are discussed.

Reproducibility: Yes

Additional Feedback: Is there any sense of how the learning rate used affect the time it takes to escape the threshold states?


Review 2

Summary and Contributions: The authors in this work study the behavior of gradient flow in the phase retrieval problem. They extend known results that show why GF can converge even in the presence of spurious local minima by investigating the behavior of the “threshold states”, i.e. states to which GF is attracted and undergo a phase transition. They empirically show that the geometry of these states changes from minima to saddles as we increase the number of samples, allowing the algorithm to escape towards the direction of the true global minimum. The authors investigate when this transition happens and they provide empirical indications for a poly-log gap between the number of samples that are known to be needed for GF to converge to the global minimum and the ones that are actually needed.

Strengths: The paper makes interesting contributions towards understanding non-convex optimization and the behavior of our optimization algorithms. There has been an increasing number of works in the last few years that help us understand the optimization landscape of various machine learning problem and this work nicely fits on that literature.

Weaknesses: The main weakness of the paper is that it provides only empirical indications for some of the claims instead of actual proofs. However, I still find the experiments well-executed and the results interesting. The authors also mention that the formula for the number of samples where the transition happens holds if we know the distribution P(\hat{y}, y). Is this a reasonable assumption? It would be good for the authors to add some clarifications regarding that. In general, I would say that the paper lacks a few explanations and intuition regarding the technical part, it is fairly hard to follow.

Correctness: The goals of the paper are well defined and the experiments are elaborate and support them well. The authors convincingly demonstrate the convergence to threshold states before transitioning towards the minimum as well as the evolution of the eigenvalues of the Hessian to characterize the phase transition. It is difficult to assess the correctness of the theoretical part though.

Clarity: Overall, the paper is well-written and well-motivated. As I mentioned before the technical part lacks some explanations and intuition to understand what’s happening. For example, in line 113: it would be useful to include the description of the theorem and some intuition about what these quantities mean and how they emerge. Some minor comments: - the phase retrieval section in page 2 ends kind of abruptly. You could mention for example why we need the Hessian, or find any way to make the transition more smooth. - in the phase retrieval section it would be better to define the number of samples as m instead of \alpha N in my opinion, since you are talking about the problem generally and you don’t have to be constrained in linear samples. You can refer to m = \alpha N, in the related work section as you do. - in line 61 you write “with perfect recovery achievable with the approximate message passing algorithm only for α > αalg = 1.13”. Do you mean that above that threshold the recovery happens in poly-time or that it is information theoretically possible? It is not clear from the context. - line 46: typo: gradient-descent flow instead of gradient flow - in Section 3: you use \mathbb{P} for the distribution while in the rest of the paper you only use $P$.

Relation to Prior Work: The paper mentions many relevant related works and has clearly stated contributions to show the extensions upon what is known. A relevant citation that the authors might want to include is “A convergence analysis of gradient descent for deep linear neural networks” by Arora et al, 2019. The authors in this work so that for linear NNs, the trajectories that gradient flow takes are such that allow convergence to a global optimum despite the fact that there are spurious local minima and saddle points in the landscape.

Reproducibility: Yes

Additional Feedback: A question that I have by looking at the graphs of Figure 1 is: is there some intuitive explanation as to why is the threshold energy higher for more samples?


Review 3

Summary and Contributions: The authors study a non-convex variant of the perceptron teacher-student learning dynamics problem, namely the so called phase retrieval problem.

Strengths: Depending on the ratio of the number of measurements over the input dimension, various dynamical phenomena are discussed in the asymptotic regime. The success in finding the global minimum by escaping critical points through unstable directions is identified with the BBP-type transition known in random matrix theory. The results provide new insights on learning dynamics and on the role of the so called threshold states. The paper is a nice interplay between analytical techniques from physics and numerics. In my opinion these techniques deserve to be better known in the ML community as they are capable of revealing quite sophisticated dynamical phenomena.

Weaknesses: The results are limited to the perceptron architecture. But going beyond this is know to be quite challenging. In the statistical physics community, the results would be considered as an application of spin glass theory. From a qualitative perspective, the results do not display any real new phenomena compared to previous studies of teacher-student learning.

Correctness: yes.

Clarity: yes.

Relation to Prior Work: yes.

Reproducibility: Yes

Additional Feedback:

[Author Response · NeurIPS 2020]

We kindly thank the reviewers (**R1**, **R2**, **R3**) for their comments. We are certain that they will improve the quality of the
work. We also thank for pointing out typos (**R2**) and missing references (**R2**), and for giving suggestions to improve the
flow of the text (**R2**) and the quality of the figures (**R1**). We will integrate these recommendations and the comments
below in the final version. We discuss the main points below.

**R1**: **Why in Fig. 2 the Hessian is not marginal before the transition?** For the size used in Fig. 2 (N=2048) the
system is already after the transition for $\alpha = 10$, see Fig. 3 and the discussion of finite size effects in Sec 2.2. According
to our results, the dynamics converges to threshold states at long time only for $\alpha$ smaller than the transition point. In
this case the spectrum of the hessian tends to a gapless one. However, the dynamics after the transition approaches only
on short times and approximately the threshold states and then aims towards the signal, which is what shown in Fig. 2
We will add in the supplementary a version of Fig.2 for a value of $\alpha$ (and $N$) before the transition where the marginality
clearly appears.

**R1**: **Mismatch between theory and simulations** As the reviewer remarked, the reasons behind the mismatch are
probably due to the finite size effects or the 1RSB approximation. Note, however, that our theory holds on timescales
not diverging with $N$—an additional mechanism that would make the system detect the signal on larger times
(polynomial in $N$), could lead to a smaller value of $\alpha_c$. We will emphasize this possibility in the conclusion.
16

**R1**: **Learning rate in the experiments.** As **R1** noted it may be possible
that a high learning rate $\eta$ shrinks the basin of attraction of minima, and
makes weakly stable states and marginal ones inaccessible to the dynamics.
Therefore this effect may shift to the right the numerical estimation of the
transition. Following the advice of **R1**, we ran experiments for $N = 1024$
and several learning rates $\eta$. The figure here shows the fraction of successes
for dynamics in 90 samples as a function of $\alpha$. We observe only a slight
shift to the right of the curve (with respect to the $\eta$ used in the paper) before
reaching a saturation. We will update the figures in the final version and add
a brief discussion.

**R2**: **Missing proofs.** The methods that we use are not rigorous but, as **R1** mentioned, are widely accepted in statistical
physics, and have been made rigorous in some specific cases. We are not aware of any proof technique for the setting
that we discuss, the formal proof is left as an open problem.

**R2**: **Knowing the distribution** $P(\hat{y}, y)$**.** Our results indicate that the algorithmic transition happens when the threshold
states undergo a BBP transition. We are left with the analysis of the Hessian of threshold states which is a random
matrix with the same form of the one analyzed in reference [26] where it was shown that the BBP threshold depends
only on $P(\hat{y}, y)$. This is reported in Theorem 1 of [26]. Therefore $P(\hat{y}, y)$ of threshold states can be used to determine
the BBP transition and consequently the algorithmic transition, as indeed we do in Sec. 2. Reducing the problem of
determining $\alpha_c$ to the knowledge of $P(\hat{y}, y)$ is a considerable advantage, since this distribution can be obtained by
replica theory, allowing us to estimate the threshold analytically.

**R2**: **The use of** $\alpha$ **N rather then m** We adopted notations used in signal processing where the relevant transitions
happen at finite $\alpha$. We will add clarifications in the text when it can give rise to confusion.

**R2**: **Comment on line 61** The threshold $\alpha_{alg} = 1.13$ is the algorithmic threshold of approximate message passing.
Above the threshold this algorithm can find the solution. The information theoretic transition instead occurs at $\alpha_{IT} = 1$.

**R2**: **Why threshold energy is higher for more samples?** The threshold states can be pictured as the states where the
dynamics converge if the labels are shuffled or substituted by random variables. In this setting when $\alpha > 1$ with high
probability there are no minima that have zero training error. Therefore the system has to satisfy $\alpha N$ constraint with $N$
variables, as $\alpha$ increases above 1 it is harder and harder to satisfy the constraints and the dynamics end up on minima
that violate more and more constraints. This implies that the energy grows with $\alpha$. We will explain this point.

**R3**: **The results are limited to perceptrons.** In this work we indeed analyzed a specific perceptron (phase retrieval)
and we expect that the same techniques could be applied to one-hidden layer neural networks with a finite number of
hidden units where some replica approaches have been already applied (e.g. Aubin et al. NeurIPS 2018). Going toward
yet more complex model is indeed an open problem.

**R3**: **There are no new phenomena compared to previous studies.** Previous work [8], that discussed very similar
phenomenology, were on the spiked matrix-tensor model that is not a supervised learning problem. The phase retrieval
we analyze is a genuine simple neural network and our work thus shows for the first time that this phenomenology
extends to this more relevant setting. Although the underlying phenomenon is the same, we remark that the techniques
used in previous works are difficult to extend to our case, and we had to develop an alternative approach.

[Meta-Review · NeurIPS 2020]

The authors in this work study the behavior of gradient flow in the so-called phase retrieval problem and postulate a phase transition in the Hessian as a function of the number of samples. The paper makes interesting contributions towards understanding non-convex optimization by studying a problem that is simple enough to allow for analytical calculations. Overall, there is a decent, well-supported agreement between theory and experiment (in particular, between the leading moments of the distribution of the threshold states as evaluated empirically and the computed moments). This paper is a valuable contribution to NeurIPS and should be accepted. Overall, however, we recommend various lines along which the paper could improve further to reach a wider audience, and we recommend that the authors revisit the author feedback before they submit their final version. First, the paper presentation is somewhat unusually difficult to follow from the perspective of the machine learning audience and could be improved by providing more background on known results that were used in the paper (e.g., the BPP transition or replica theory), if necessary in the appendix. Also, moving some mathematical results to the appendix for the sake of creating space for a non-technical discussion on the relevant background in the beginning of section 2 would make the paper more readable. Also, descriptions on the implications and meanings of the theoretical results that were used (e.g., in line 113) should be provided. Second, it would be nice if the authors could discuss the possible origins (i.e., the validity of assumptions made) of the quantitative disagreement between their theoretical prediction of the threshold state and their experiments, if possible.